# Horticultural Therapy for Individuals Coping with Dementia: Practice Recommendations Informed by Related Non-Pharmacological Interventions

**DOI:** 10.3390/healthcare12080832

**Published:** 2024-04-15

**Authors:** Matthew J. Wichrowski, Monica Moscovici

**Affiliations:** 1Rusk Rehabilitation, NYU Grossman School of Medicine, NYU Langone Health, New York, NY 10016, USA; 2NYU Langone Health, New York, NY 10016, USA; monica.moscovici@nyulangone.org

**Keywords:** nature-based therapy, horticultural therapy, complementary therapies, dementia care, Montessori methods

## Abstract

Dementia care currently presents a challenge to healthcare providers on many levels. The rapid increase in the number of people with dementia and the costs of care certainly contribute to these challenges. However, managing the behavioral and psychological symptoms of dementia (BPSDs) has become one of the most significant tasks in providing care and can lead to poor health and well-being outcomes, not only for the people living with dementia (PLWD) but also for those providing their care. Cost-effective, easily implemented, highly adaptable, empirically based alternatives are needed. Interventions such as Horticultural Therapy (HT), which is naturally informed by Montessori-Based Methods for Dementia and sensory reminiscence therapies, meets these qualifying factors. This article, based on a review of current best practices and clinical experience, hopes to provide recommendations for such an intervention along with special considerations for PLWD and adaptations for different acuity levels. With additional safe and effective, person-centered, non-pharmacological interventions available for the complex cognitive and neuropsychiatric manifestations of this disease, a better care milieu can be provided, improving the quality of life for both patients and caregivers. This article also identifies the need for continued research into the synergistic effects of person-centered behavioral and psychosocial interventions combined with environmental approaches to provide the optimal healing environment for those coping with dementia.

## 1. Introduction

Currently, more than 55 million people are living with dementia (PLWD) worldwide, and almost 10 million new cases are being diagnosed every year. The global economic cost of dementia reached 1.3 trillion (US dollars) in 2019, trickling down to healthcare facilities and personal costs [1]. Considering these numbers as well as the American baby boomer population, set to begin aging into the dementia range, [2,3], the amount of PLWD (and their ensuing costs) is predicted to triple by 2050 [4]. This exponential rise in elderly individuals with dementia will require an equal increase in more specialized care. Memory care facilities can achieve such care by integrating newer, cost-effective, non-pharmacological, and person-centered alternatives to managing the complex and increasing cases of dementia [2]. 

Dementia is a general term that encompasses the “loss of memory and other mental abilities severe enough to interfere with daily life” [5]. Such losses are brought on by physical changes in the brain due to a multitude of causes ranging from diseases such as alcoholism and Huntington’s disease to traumatic brain injuries and family genetics. Although Alzheimer’s is the most common type of dementia, there are four other major categories of dementia. These include Vascular Dementia, Frontal-Temporal Dementia, Lewy Body/Parkinson’s Dementia, and Mixed-Type Dementia. Over 120 causes, variations, and types of dementia have been identified thus far [5]. Even though they can all be defined by the decline in cognitive function, people living with all types of dementia also “experience a host of neuropsychiatric symptoms” [6]. Although most often studied in patients with Alzheimer’s disease, the resulting behavioral and psychological symptoms of dementia (BPSD) can complicate any cause of dementia. These symptoms are among the most distressing to both patients with dementia and their caregivers. They are also associated with a high economic burden [7]. Such symptomatology can also become an underlying cause of common adverse outcomes for PLWD. These possible outcomes can include “increased hospitalization, decreased quality of life, and increased distress among patients and caregivers, as well as being associated with more rapid progression of disease” [6,8]. This article aims to address two BPSDs that can be especially difficult for daily caregivers. Agitation and depression affect emotional, social, and ultimately physical domains. They are also the symptoms that are most proven to be alleviated by Horticultural Therapy [8]. 

The neuropsychological symptoms of depression and agitation tend to cause social isolation and/or low self-worth with a lack of motivation. This amalgamation of symptoms and their affected domains causes a highly decreased quality of life. Not only does this affect the person with the disease but also their family, friends, and caretakers [6].

Caretaker well-being is an important consideration in this equation. Formal caregiver training and ability has been proven to be the most effective BPSD intervention [9]. Trained caregivers are more likely to provide “high-intensity care” and handle behavioral problems. They also report higher amounts of strain, mental and physical health problems, and caregiver burnout [10]. The psychological well-being of a dementia caregiver can also affect their care recipient. This often stems from the negative aspects of caregiving which in turn increases the chances of depression and anxiety in their care recipient [11]. The CDC estimates that about 60% of dementia caregivers suffer from high rates of emotional distress and approximately 40% report symptoms of depression [12]. 

Considering these factors, the alleviation of BPSDs becomes highly important for PLWD and their immediate circle, especially caretakers, affecting the level of care they can provide [13,14]. Identifying effective neuropsychiatric symptom management techniques in PLWD, benefiting both patient and caregiver well-being, has become an important clinical goal. To that end, this article aims to offer helpful information for staff, caretakers, and activity providers.

Using non-pharmacological means can provide both physical and emotional symptom alleviation [15]. These interventions are often preferred due to the possibility of medications precipitating neurobehavioral disturbances [6,16]. Proven alternatives that help with BPSDs include Montessori-Based Practices, the agency of choice/motor independence, and sensory reminiscence therapy [16,17,18,19,20,21,22,23,24,25,26,27,28,29,30,31,32]. These practices and their benefits naturally inform the provision of Horticultural Therapy (HT). HT is defined as the use of gardening activities for therapeutic exercise aimed at meeting individual treatment goals [33]. Studies from a systematic review utilizing HT and sensory gardens in caring for patients with dementia showed significant improvements in agitation levels as well as time spent engaged in activity. These results show encouraging possibilities for administrators in care facilities to use HT in caring for patients with dementia [4]. Utilizing such interventions can provide a clinically relevant, person-centered approach to meet the needs of increasing populations of persons coping with dementia and their caretakers.

## 2. Aim

The aim of this article is to explore contributions from the field of horticultural therapy as informed by associated non-pharmacological approaches to explore current aspects of dementia care, and in particular, to offer design and practice recommendations specifically tailored to the needs and clinical goals of those coping with dementia in its various forms.

A search and analysis of both the relevant classical and current literature was performed. With this, in combination with clinical expertise, the authors hope to integrate findings to offer safe, cost-effective, person-centered intervention options for BPSD management in patients and additionally reducing caretaker burden.

## 3. Methods: Data Sources and Search Strategies

Google Scholar and Pub Med were searched with the keywords: ‘sensory gardens and dementia’ (6650 results), ‘gardens and dementia’ (17,000 results), ‘dementia and behavioral challenges’ (17,000 results) and ‘horticultural therapy and dementia’ (3020 results). Articles were assessed for a date range of ‘since 2020′. Articles were reviewed and discussed by authors. Criteria for selection included meta-analysis, review articles, RCTs, quantitative studies having high ‘n’ and utilizing validated scales, and mixed-method studies with high ‘n’ and validated scales. Consideration of earlier studies included foundational works and “classic studies”. Further citations were derived from mining the reference lists from previous articles. Lastly, articles addressing unique approaches and interventions were explored for additional unique contributions to patient care according to the aims of the paper. Utilizing open-access journal articles was a key consideration in the literature search to enable wider ease of access. The large number of results precluded a systematic review, so the above-mentioned criteria combined with clinical expertise and author discussion guided the reference selections. Fifty-nine articles were chosen to inform the recommendations in this article.

## 4. Results: Evaluating Practices Supporting Person-Centered BPSD Management 

The management of BPSDs is a highly individualized process. Following a thorough assessment by appropriate medical professionals, it is important to address any possible contributing medical problems or medication effects [9]. Concerns regarding the use of pharmacological therapies are addressed. In addition, non-pharmacological approaches shown to impact behavioral symptoms in dementia care are explored.

### 4.1. Pharmacological Therapy

Patients with dementia often are prescribed multiple medications. While pharmaceuticals that address medical problems and pain associated with dementia are important in managing BPSDs, there is evidence that some of these medications may be contributing to BPSDs. The reduction or discontinuation of medications associated with increases in BPSDs, as approved by medical personnel, could significantly improve BPSDs and associated difficulties [9]. For example, a recent study showed that “reducing anticholinergic burden by at least 20% significantly reduced the severity and frequency of BPSD and decreased caregiver burden” [34]. Drug–drug interactions should also be considered when reviewing or making changes to medications. 

While psychiatric medications can help ease BPSDs, side effects can be just as problematic as the symptoms they are designed to combat [35,36]. While polypharmacy increases the possibility of negative side effects, it may even cause more serious issues “such as falls, strokes, or death” [6,36]. Due to such issues, pharmacological interventions should only be prescribed when “behaviors pose a significant safety risk” or if the person coping with dementia is extremely distressed. Although there is a lack of strong clinical evidence concerning the research and implementation of non-pharmacological interventions, the lack of adverse effects makes them promising options for managing BPSDs in situations that are not imminently dangerous. These interventions should include patients as well as their formal caregivers for best outcomes [9].

### 4.2. Horticultural Therapy

E.O. Wilson’s Biophilia hypothesis states there is an innate biological and evolutionary connection between humans and nature [37] which has significant effects on mood. The findings of a recent meta-analysis indicate “that exposure to natural environments had a medium to large effect on both increasing positive affect and decreasing negative affect” [38]. It has been shown that human beings have a natural fascination and attraction to certain qualities of nature. 

Another important foundational theory in HT work is Attention Restoration Theory (ART). When environments contain the features of being away, fascination, compatibility, and extent, it allows for effortless attention to nature that can offer rest and restoration from mental fatigue [39]. The ability to pay attention to one’s environment and focus on one’s goals is an important function in life. HT has the potential to impact many domains of function and can provide mental, physical, cognitive, emotional, social, and spiritual benefits. 

The American Horticultural Therapy Association (AHTA) defines HT as an activity-based process that is part of a predetermined treatment plan in which “the process itself is considered the therapeutic activity rather than the end product”. This type of nature-based therapy can be utilized as part of a stand-alone or team treatment approach. Therapeutic Horticulture (TH) is a closely aligned practice that aims to improve well-being through active or passive involvement with nature. TH uses the “restorative value of nature to provide an environment conducive to mental repose, stress-reduction, emotional recovery, and the enhancement of mental and physical energy”. A TH session focuses on the psychological, physical, and social needs of its participants [33]. 

These activities (HT and TH) can be easily implemented in a variety of settings. Interventions can range from adjusting and enhancing an already existing garden (specializing its features to meet the needs of the particular populations it serves while adding biophilic features) to hands-on indoor planting or sensory-based herbal craft activities. These activities can further be graded to reflect the ability level of the participants and provide an appropriate level of challenge to assist in meeting their clinical goals. Specific areas in physical, emotional, cognitive, and social domains can be targeted [40].

Although a relatively new field, HT can provide benefits for a wide range of different populations. This can include, but is not limited to, veterans; people with special needs, psychiatric disorders, and experience of substance abuse; those in skilled nursing and assisted living settings, cancer (both active and remission) patients; those in hospices/palliative care; and homeless populations [33]. HT can also be immensely relevant to PLWD. Cognitively, the use of HT/TH has been proven to stimulate memory [41] and improve attention capacity [42]. Guided horticultural activities can also serve to exercise sequencing abilities, generalizable to Activities of Daily Living (ADLs). Psychologically, it has been shown to improve mood [43] as well as quality of life [44], while alleviating depression [45]. HT/TH can improve personal worth [46], increase self-esteem [47], and enable a sense of pride and accomplishment. Lastly, it has even been shown to provide an increased sense of stability [48]. 

The above benefits are incredibly useful to PLWD and offer great possibilities for the improvement of BPSDs. In fact, the effects of sensory gardens and HT programs on improvements in agitation levels and time spent engaged in activity are well documented for PLWD. Not only is engagement shown to increase but the overall time spent in inactivity is shown to decrease when HT or sensory gardens are introduced. Even personal affect became more positive during HT when compared to traditional activities [4]. Planting and other nature-based activities, whether inside or outside, can promote a range of benefits, including the provision of a non-threatening, mood-enhancing activity that promotes social opportunity, inclusion, and support.

### 4.3. Montessori Method for Dementia

HT is naturally informed by the Montessori Method for treating dementia, which has been developing for the past twenty years and is used on an international scale. Multiple reviews have reported that Montessori-Based Practices (MBPs) improved constructive engagement and had positive affects [16,17,18], as well as improved eating behaviors and cognition [14]. The method typically involves “(a) identifying an activity of interest that is reflective of the individual’s skill level; (b) making use of familiar materials and objects; (c) breaking the activity down into small steps; and (d) inviting the individual to complete the task themselves” [18]. This process can increase the meaningfulness of the activity and improve motivation and participation.

The Montessori Method relies heavily on the agency of choice and motor independence, another proven alternative to the use of pharmaceuticals to help treat BPSDs. “As human beings… we need to be proud of our abilities and feel respected by other people. These experiences can often be hard to achieve for persons with dementia, partly due to symptoms of dementia and also due to other persons’ reactions” [19]. This often results in feelings of “failure/weakness, losing face, ignorance, and conflict” [20]. Therefore, a feeling of agency and independence becomes very important.

However, because PLWD can have inaccurate perceptions of their abilities, it is practical to provide both the opportunity to use their abilities while also getting help to remember how they previously used these abilities [19]. The solution to this balance between independence and support/safety is to do things together—as in doing things with the person rather than for them, focusing on the things the person can do, rather than the things they cannot—while offering reassurance, encouragement, and plenty of time and patience, if needed [21].

It is important for the participant to be involved as much as possible in each step of the activity, with the focus being on maximizing individual contributions during the process rather than on the product. This can simply mean enabling a person to do things their way, within reason [22], while also offering a potent tool for therapeutic improvement. A comparison of behaviors and speech, both before and after utilizing this method, showed that PLWD displayed a decrease in the negative themes that come with perceived inability and its BPSDs, as well as an actual increase in ability [23].

### 4.4. Sensory and Reminiscence Therapy

Sensory stimulation refers to a variety of techniques and activities used to exercise the senses, helping to increase alertness and reduce agitation [24]. It includes visual, olfactory, tactile, gustatory, auditory, and even kinesthetic stimulation [25]. Offering the opportunity to explore stimuli can bring a state of pleasant relaxation [26]. The addition of nature-based stimuli has the potential to further increase the benefits. Studies show that “simply touching wood or viewing an image of roses for just three minutes can induce beneficial physiological responses” including relaxation and stress reduction [49]. Because nature allows for effortless attention and connection [37,39], it offers easily attainable benefits with less work for both patient and caregiver. 

The beneficial effects of sensory stimulation can extend beyond relaxation (both physical and mental) to opportunities for returning to a sense of self. According to some dementia literature, “cognitive impairment in people with dementia is often regarded as representing a loss of self-hood” [19,29]. This view implies the separation of the mind from the body. The significance of embodiment for PLWD (and all people in general) cannot be stressed enough [27]. People living with dementia tend to dissociate from their mind, body, and current moment in time [19]. The grounding action of embodiment not only helps release stress and improve mental well-being [28] but can offer some reprieve from symptoms of dementia [24]. Sensory stimulation therapy offers the opportunity to reorient to the present by producing increased mental alertness through sensory stimulation. At the same time, it has the potential to improve quality of care by expanding on the caretakers’ communication methods for connecting with patients suffering from BPSDs [29]. Therapeutic benefits can be enhanced through the addition of reminiscence therapy, which typically ties to the sense of smell but can also include the other senses and even muscle memory. 

For over 50 years, memory stimulation in the form of reminiscence therapy has been seen to help process life events and prepare for death, with added benefits of mental and psychosocial well-being [30,31]. Knowledge and understanding of a patient’s personal history and preferences can contribute significantly to optimal patient care and patient satisfaction [30]. Both knowledge of history, through the eyes of the patient, along with information gained from sensory stimulation allows caretakers to view patients as individuals and find new meaning in their behaviors that push past just being symptoms of a disease and more towards a way of relating, especially if there is an unmet need [29]. Reminiscence, life review, cognitive stimulation, music therapy, and aromatherapy, among others, have demonstrated positive effects on people living with dementia [32] and their caretakers [29]. While these therapies were in use, such as music played during baths, multisensory stimulation room sessions, and aromatherapy or massage, BPSDs were greatly improved. Communication, quality of life, and function also improved [27]. As a result of such improvements, the staff members experienced equal improvement, with a sense of excitement that made them even more willing to implement such interventions [29].

## 5. Variations of Horticultural Therapy Implementation

### 5.1. Doing and Being in the Garden

There are a multitude of ways in which HT can be implemented, depending on team treatment goals, contraindications, acuity level, and participant preference. When implementing HT, whether indoors or in a specially designed garden, it is helpful to consider both passive benefits as well as specific prescribed activity as a therapeutic exercise designed to meet the clinical needs or treatment goals of participants. The combination of setting and activity creates the potential for a wide range of therapeutic benefits for an equally wide range of abilities [6].

### 5.2. Memory Gardens—Setting Up for Being in the Garden

Memory Gardens can be a great therapeutic horticulture option. To be effective, they should contain several elements while avoiding elements possibly hazardous to people with dementia (Table 1).

Having a garden that allows visitors to be able to orient themselves through visual cues and a path that brings them back to where they started provides increased feelings of control, self-confidence, and reinforcement to go outdoors [46]. The ability to go outdoors, especially alone (when assessed by a clinician or caregiver based on current weather conditions and the affect/ability of the individual), has been shown to alleviate the agitation and depression symptoms of dementia [53]. Just being around plants can help enhance that effect, offering a soothing, quieting quality [6,48,54]. Providing a space for gathering, sheltered from the sun, can offer opportunities for socialization, another characteristic that can improve such symptoms [50,51,52]. This mix of structured and unstructured activity is a key benefit to garden visitors as they can be afforded the autonomy to move about as they see fit while providing a degree of direction. The garden setting can also provide a purposeful way to meet ambulatory goals. A well-designed garden enhances enjoyment, increases utilization, and creates a rich setting to employ HT activities that further enhance the therapeutic potential for participants. This effective combination magnifies the overall experience of horticultural therapy [6].

Including one or two sturdy garden beds that PLWD can sit, stand, or pull their wheelchair up to can be a great feature for annual plantings such as herbs, geraniums, or other familiar flowers. Most people tend to have some past personal experience with or memory of gardening, especially with parents or grandparents. This is great for stimulating reminiscence. The fragrant olfactory stimulation offered by herbs is also recommended in any memory garden. Benches (facing the path) and railings, at appropriate intervals, are also recommended to allow garden visitors to rest or gain a little extra help while still feeling independent [50,51,52]. If no easily reachable outdoor space is available, an enclosed balcony or sunroom with a garden bed that PLWD can stand, sit, or bring their wheelchair up to can also achieve similar benefits. The use of low-cost LED lighting can embellish existing light in a darker space and create a warm-feeling, plant-friendly space for participants to enjoy, especially in winter when conditions outside may not be conducive for garden visits. Settings with hot summers will also benefit from a climate-controlled indoor garden.

### 5.3. Planting Activity Example—Setting Up for Doing in (and out of) the Garden 

The addition of prescribed garden activities aimed at meeting the clinical and personal goals of the individual adds a range of benefits for participants. Being able to learn about and explore a few plants (usually no more than three depending on cognitive status) offers cognitive stimulation and the opportunity to reminisce and engage with texture, thickness, and different color variations while smelling and feeling the soil or the plant itself. This provides the opportunity to exercise most of the senses. Choosing one’s plant also provides some personal control and choice during the activity and an increased sense of meaning. 

The simple act of transplanting a starter plant into a larger pot, to give it more room to grow, offers metaphorical opportunities for discussion which can positively impact self-confidence and motivation [32,46,47,48]. This can also be expanded to propagation and seed starting activities as well, for extra variety, keeping in mind that simple steps, adapted for different acuity levels (Table 2), done one at a time, while offering two or three choices for each step tend to work well.

Such activities set up the conditions for easy, sequential task completion, allowing opportunities for independent tiered goal achievement. Sequencing exercises performed during an enjoyable, structured task may generalize to other important tasks requiring sequencing such as ADL skills [40]. Allowing participants to have a hand in every single step, even if with assistance, elicits a feeling of accomplishment and pride (Table 3) [19,21,41]. 

Having something to take care of something (i.e., the finished plant project)—in a situation where people have little control over anything and others are doing everything for them—offers a feeling of control and usefulness in the world [44,46]. Not only does this bring cognitive, emotional, and psychosocial benefits, but it also enables the use of gross and fine motor skills. If there are standing tolerance goals, the planting activity can also be done in conjunction with other rehabilitation therapies.

### 5.4. Soil-Less Activity Example for Immunological Concerns

Planting and soil activities may not be appropriate for everyone. For example, some individuals prefer not to have any interaction with soil, even if wearing gloves. In some such cases, planting activities may become counterintuitive to the alleviation of agitation. There may also be situations in which soil is contraindicated due to specific immunological and infection concerns. These considerations and preferences should be assessed and evaluated by the medical team and activity staff, respectively. 

While planting offers many opportunities for sensory practice, other activities such as an herbal aromatherapy sleep pouch activity can provide additional opportunities to exercise the senses while experiencing nature. Such activities can start with an exploration of different dried herbs while learning about their benefits. Next comes comparing their colors and shapes while feeling their textures and squeezing the herbs between a participant’s palms or fingers. Listening to the noise they make when rubbed, and finally, smelling their released essential oils while reminiscing about memories can provide a powerful sensory experience [56].

Providing information about the benefits of the herbs, plus information about their historic uses at the start of the activity, offers cognitive and auditory stimulation. Offering three different herbs that are known to help with different parts of sleep [57,58] (like lavender, lemon balm, and rosemary) allows for the agency of choice, providing participants the option of refusing to use or interact with specific plant materials. Some people have very good reactions to certain smells, while others have negative reactions to the same smell. Exercising such preference in the agency of choice can be easily built into the activity.

This activity also offers an opportunity for the use of fine motor skills by squeezing and rubbing the herbs to release their essential oils (scent). Mixing and pouring the herbal blend into the pouch also offers gross motor stimulation, and so does ripping up tissues to fill the pouch and help distribute the herbs more evenly. Finally, squeezing and mixing the pouch once closed, to further distribute the materials inside and release their scent, allows for the use of fine finger motions. Completing such a project and putting it in between their pillow and pillow case can bring participants a sense of accomplishment and some ownership over their well-being. Lastly, this activity is not limited to immunocompromised participants alone. It can be used for the general dementia care population and can be adjusted to a spice blend sachet or herbal tea blend, with proper participant, nutritional, and safety considerations, by using a paper coffee filter and 6-inch cotton string, instead of a pre-made muslin or cotton pouch. 

## 6. Special Considerations for People Living with Dementia

### 6.1. Memory Gardens—Being in the Garden

Montessori Methods for Dementia are known for focusing not only on supporting the individual but also on using and adapting the environment to provide safe yet independent engagement. The environment itself can be used as the support to promote independence. “We are unable to change the devastating effects of dementia but we can incorporate strategies and alter the environment while providing meaning and purpose to the day—so that the person not only engages in life, but has the opportunity to maintain, and even restore function” [22]. Memory gardens are the perfect example of this. On average, more than 90% of residents stay inside their unit in a lying down or sitting position throughout the day [53]. Offering enticing, supportive, and restoring environments can offer important interventions for such common issues.

Norton, 2023, has provided robust support from multiple studies that show a range of positive findings for therapeutic garden interventions when applied to help the dementia population. The majority of these findings were related to an increase in overall quality of life but there were many other benefits in individuals receiving this therapy, including a “reduction in agitation, positive changes in behavior, physical and mental health benefits, alleviation of social isolation, and potential changes in cortisol levels” [2].

### 6.2. HT Activities—Doing in (and out of) the Garden

Through this author’s experience of providing a variety of HT activities for PLWD, a few things have proven themselves to work well. It may seem redundant, but introducing oneself, while pointing to a nametag—so participants can see where to find your name—and connecting yourself to something—such as holding up a plant—at the beginning of every HT activity is highly beneficial. So is a short introduction of what is about to happen and why. People in later stages of dementia often forget why they are there. Starting an activity without any introduction can be jarring under such circumstances. Simple transitions allowing for extra cognitive preparation and an understanding of what is expected from them during the activity can help alleviate this type of common discomfort. This is an element that can be easily forgotten by activity facilitators, especially if there is a mixed group of participants in varying stages of dementia, but such extra transition and preparation opportunities can be useful to ensure successful activity outcomes (Table 4) [29].

Safety concerns include areas of mental, emotional, and sensory safety. Some participants tend to have sensitivities to lights, sounds, and temperature. Sunglasses, earplugs or strategic microphone speakers, blankets, seating charts, and lots of support, encouragement, and patience for participants’ independence can all address such sensitivities and allow for safe and successful sessions on all fronts. Likewise, working effectively with alternative behaviors maximizes safety and comfort during the activity and success in achieving HT goals (Table 5).

Lastly, PLWD are not always oriented to the present reality, but we can live in a shared reality while working on an activity together. This offers a therapeutic calm and stability for the activity participants and can also offer an opportunity for the formation of trust and bonds between the participants and caretakers. Allowing participants to feel comfortable and safe by not contradicting their current reality builds trust and can offer opportunities to see flickers of their personality and history. This can help caretakers understand and adapt better to individual needs and proclivities, while helping various activities of daily living proceed smoothly and providing a more positive work and therapeutic environment [23,25,26].

## 7. Conclusions

Dementia care is moving towards a more multi-disciplinary, person-centered approach. However, with the predicted exponential rise in the number of dementia populations, an equal rise in more specialized care will become necessary [2,4]. It will benefit healthcare providers to begin exploring new options for the management of behaviors that are cost-effective, easily implemented, highly adaptable, and empirically based. The use of HT, including specifically designed therapeutic gardens guided by Montessori-Based Practices fulfills these criteria. The design and activity recommendations offered provide opportunities to reduce BPSDs as well as improve physical, cognitive, emotional, and social function. Along with increased engagement and decreased inactivity studies in a systematic review of HT use in caring for patients with dementia, there were decreased scores on the Cohen-Mansfield Agitation Inventory (CMAI). Such evidence should encourage administrators across care facilities to include HT interventions in their care milieu. To summarize, HT is a suitable, stimulating, and productive activity for dementia care programs. 

Given the growing use of complementary therapies for promoting well-being, there is a need to study new therapeutic approaches that are available in order to find ways to ensure a high quality of care. This will help define and disseminate best practices for current and future PLWD across care settings [27] to provide the highest quality of life possible for those coping with dementia. Further research should continue to explore the relative benefits of complementary practices in dementia care including interventions such as HT alone or informed by other accepted practices. Sometimes the right mix of setting, approach, and activity can combine to have a robust therapeutic impact on the quality of life of those impacted by dementia and those who provide care for them.

## 8. Limitations and Future Directions

Although informed by a thorough review of the literature concerning the topics addressed, this paper is not meant to be a systematic review. Criteria for the literature chosen was based on the level of evidence with preference given to recent meta-analyses, review articles, randomized control trials, and higher ‘n’ studies utilizing validated scales to provide background and support for practice recommendations.

There are implicit limitations in this approach as compared to more systematic explorations. Yet, considering the aims of this paper, a wide array of current and classic resources provided information to back the treatment practices recommended herein. The recommendations provided are also informed by the successful clinical practice and experience of the authors. While there are limitations associated with this approach, it provides a sound foundation for patient-centered dementia care, with implications for exploring combinations of both environmental and clinical approaches to address relevant challenges and provide optimal care for those coping with this condition. There is potential in utilizing horticultural therapy and therapeutic garden design approaches to explore the efficacy of various interventions and combinations of interventions to maximize quality of life.

This exploration offers promise in developing a research agenda to further assess the effects of nature-based interventions in dementia care. In particular, nature-based interventions are well suited to address behavioral challenges and other clinically relevant issues such as supporting mood and providing cognitive and social stimulation to impact quality of life for both patient and caregiver. Effective combinations of unique interventions with synergistic effects are worthy of exploration in an experimental context.

## Figures and Tables

**Table 1 healthcare-12-00832-t001:** Recommended garden elements and their precautions [50,51,52].

Element	Considerations	Precautions
Boundary	An enclosed space is important for safety It is equally important for visitors not to feel confined so screening fences and walls with hanging plants can minimize feelings of being closed in.	Using locks on all gate apparatuses or openings in between fences is important since PLWD may be able to open regular gates and divisions in fences that are not meant to be opened
Walkway	A paved path with a contrasting color on each side/edge to help delineate safe, flat walking areas from uneven garden/planted areas should be provided This path should be wide, solid, and safe enough to support wheelchair access The path should form a loop that begins and ends in the same location, with focal points that provide interest along the way and signs that clearly mark an extra wide entry/exit and other important points Interpretive signage can add cognitive stimulation and provide a sense of meaning to enhance the garden experience	Raised curbs or edges along walkways can cause a tripping hazard Walkways that are not level are discouraged for similar fall risk issues
Planting Considerations	Plants that engage all of the senses offer many opportunities for stimulation including: ○ fuzzy and/or textured plants; ○ fragrant plants; ○ edible plants (i.e., mint and other culinary herbs). Plants that are most likely to be familiar to garden visitors such as geraniums and roses Something of interest for every season (i.e., a mix of annuals and different perennials)	Plants with any kind of toxicity level are discouraged Plants with thorns, spikes, or sharp edges, like cacti, lemongrass, certain roses, and bromeliads are discouraged Plants that have a chance of causing contact dermatitis are discouraged Plants whose pollen causes common allergic reactions (i.e., chamomile is in the ragweed family and may cause a reaction for people allergic to ragweed pollen) are discouraged

**Table 2 healthcare-12-00832-t002:** Recommended adaptations for acuity levels when working with soil [40,55].

Level of Acuity	Adaptation	Sequence of Steps
Higher functionality—ability to stand	Use soil placed in a big, wide container/bucket with high sides placed on a table or surface where participants can stably stand and mix	1. Mist/spray soil with water; 2. Mix soil; 3. Mist/spray perlite with water; 4. Mix perlite; 5. Then, mix perlite into soil (participants are usually good at choosing proper ratios on their own so have them decide what looks right); 6. Fill the empty pot one-third with soil; 7. Squeeze the pot with the plant inside to release roots; 8. Turn the pot upside down while holding the top of the soil between flat open fingers falling in between the plant; 9. Release the plant and place it right-side up into its new pot; 10. Fill all the empty spaces around the plant with soil.
Higher functionality—wheelchair	Use soil placed in a container with lower sides so participants who are sitting down can see inside	
Medium functionality	Use soil placed in a closed Ziplock bag for easier accessibility than in containers and more opportunity for finger mobility practice when mixing/squeezing with bare hands This is an especially good option for patients who are reluctant to touch the soil or get their hands dirty	
Medium functionality—less tendency to eat materials	Use measuring cups to add different amounts of perlite to the soil or to scoop the soil into pots This can add some level of familiarity, especially if the participant was not a gardener	
Lower functionality and/or tendency to eat materials	Use soil and perlite in closed Ziplock bags only Move all materials away from the participant after completing each step Work in a team or assembly line fashion	Keep the soil in a Ziplock bag and in the hands of an activity provider at all times; Present the opened bag to allow one participant to mist the soil; Close the Ziplock bag and have the next participant mix the soil in the closed bag; With the Ziplock bag open in the hands of an activity provider, have one or two people take turns filling the new pot under supervision; Allow the next participant to squeeze the pot with the plant inside to loosen its roots; Have another participant turn the plant, along with its pot, upside down to release the plant from its pot and “tickle” its roots to spread them out a bit Have one participant make a hole in the pot from step 4 that is big enough for the plant and place the plant in the pot; Allow the next participant to fill in the soil around the plant and pat it down to “tuck in the plant” and stabilize the roots.
For all Acuity Levels	All equipment and materials used (e.g., trowels, plastic soil containers, bags and bins, water sprayer handles, pouches, seeds and seed containers/shakers, etc.) should be larger in size than normal for ease of use in this population and to prevent client frustration and feelings of helplessness while participating in the activity.	

**Table 3 healthcare-12-00832-t003:** Recommendations for types of acuity-based assistance [40,55].

Level of Function	Assistance Strategy for Activity Providers/Caregivers
High	Use mostly verbal cues with minimal physical assistance
Medium	Ask the participant to do part of the task For example: ○ Hold the pot while you fill it with soil or vice versa; ○ Hold dried herbs in an open hand while you use one hand to rub them or hold them in a flat open hand while you direct and tip them into the pouch; ○ Hold and pull one side of the string or ribbon when making a knot to close the pouch.
Low	Hand over hand, where your hands are over theirs, makes the activity easier Or having the participant’s hand over yours, squeezing when they want you to: ○ Cut (with a scissor while pruning); ○ Squeeze (the pot you are holding to loosen the roots); ○ Write a letter on a plant label; ○ Push the label into a predetermined spot.
Low functionality	Total hand-over-hand physical assistance Or demonstration while participant chooses options by where their gaze lands

**Table 4 healthcare-12-00832-t004:** Recommended material considerations.

Material	Special Considerations
Soil	Use OMRI-certified organic, indoor potting mix as the planting medium
Perlite	Add perlite to lighten the soil mix due to general tendencies of overwatering (a little too dry is better than too wet)
Tools	Use plastic hand trowels and rakes as opposed to metal or wood Use safety scissors when scissors are a necessity Keep track of all tools at all times, especially at the end of the session
Any other added craft material or plant decoration	Use materials larger than 1 ½” to lower the possibility of choking [55]
Aromatherapeutic Herbs	Avoid: ○ Common allergenic plants such as those in the ragweed family (chamomile, mugwort); ○ Finely ground herbs such as marjoram; ○ Things that make one sneeze, such as hot pepper or other spices [59]; ○ Ingestion of Lemon Balm for those on sedatives or hypothyroid medication [56].

**Table 5 healthcare-12-00832-t005:** Recommendations for working with alternative behaviors.

Behavior	Recommendations for Activity Providers/Caregivers
Dismantling projects	Take the project back after each step or move it away from the participant Make it a team/assembly line project
Short attention span/always fidgeting or doing things that may be unhelpful to the activity or group	Provide reading/picture material as it coincides with the project while explaining, demonstrating, or working with everyone else Provide small coloring, word search, or other printed games while everyone else is engaged in listening but make sure the coloring page or game is simple so as not to take too much time, losing their activity participation
Eating activity materials	Make it a team/assembly line project Put whatever materials you can in Ziplock bags (i.e., soil) Use additional craft or plant decoration materials larger than 1 ½” to eliminate choking hazards
Yelling/inappropriate remarks	This can sometimes mean there is an unmet need so asking caretakers who know the participant best can let you know if everything is ok Sometimes replying with an appropriate acknowledgment or word of conversation can help In some cases, when all else fails, not paying attention to the behavior may be the only option.
Disagreements between participants	Ask care staff to help seat participants according to who they get along with best and place persons that often have disagreements farthest apart De-escalating situations with jokes or a tactful redirect back to the activity
Refusing to participate while being present	Sometimes “participants” like to watch and just being present and out of their room is enough to meet their goals Asking the “participant” for simple opinions on materials or placement can encourage a type of activity engagement Invite the “participant” to be your assistant for the activity or for set up/clean up

## Data Availability

Not applicable.

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
