# Peer review of "Horticultural Therapy for Individuals Coping with Dementia: Practice Recommendations Informed by Related Non-Pharmacological Interventions"

_healthcare, 2024, doi:10.3390/healthcare12080832_

Round 1

Reviewer 1 Report (New Reviewer)

Comments and Suggestions for Authors

Line 26-89 Introduction

There are various types of non-pharmacological interventions available for patients with dementia. In the introduction, it appears necessary to present a fundamental reason for recommending horticultural therapy interventions by introducing various non-pharmacological interventions and mentioning their pros and cons. Additionally, to my knowledge, horticultural therapy is known to be more costly compared to other treatment methods in terms of economics. Hence, the reasons for choosing horticultural therapy over other low-cost treatment methods need to be emphasized further. This could be adequately explained with various prior studies.

Line 102-112  3. Methods: Data Sources and Search Strategies

The research method section of the paper should be detailed enough to allow other researchers to replicate the study. However, based on the current description, I believe the replication of the study is unfeasible. Although this research is not a systematic review, it resembles the first strategy used in systematic reviews in terms of search strategies. Specifically, if numerous studies were initially obtained through a search, a detailed explanation is needed on the criteria used to select and exclude studies for organizing the research results.

Line 118-140  4.1. Pharmacological Therapy

There is considerable evidence recommending the concurrent use of pharmacological and non-pharmacological treatments for dementia patients. However, generalizing the severe side effects reported in some cases of drug therapy to all dementia patients seems imprudent, and the benefits of combining drug and non-drug treatments appear to be superior. This perspective should also be addressed in this section.

Line 141-259 

I believe each therapy is well summarized.

Line 261-413

I think the main core part of this research is well organized. Especially, the sections that are very useful for practical application when actually applying the therapy are impressively well-compiled.

Overall, the organization of the results is commendable; however, procedural transparency is also necessary for the paper to hold value as a scholarly article. Therefore, I believe that if the research methodology section is elaborated upon, this study would possess significant value as a publication.

Author Response

Reviewer 1

Line 102-112  3. Methods: Data Sources and Search Strategies

The research method section of the paper should be detailed enough to allow other researchers to replicate the study. However, based on the current description, I believe the replication of the study is unfeasible. Although this research is not a systematic review, it resembles the first strategy used in systematic reviews in terms of search strategies. Specifically, if numerous studies were initially obtained through a search, a detailed explanation is needed on the criteria used to select and exclude studies for organizing the research results. Methodology sect presented in more detail with article selection process described

Line 118-140  4.1. Pharmacological Therapy

There is considerable evidence recommending the concurrent use of pharmacological and non-pharmacological treatments for dementia patients. However, generalizing the severe side effects reported in some cases of drug therapy to all dementia patients seems imprudent, and the benefits of combining drug and non-drug treatments appear to be superior. This perspective should also be addressed in this section. Adjusted according to reviewer comments.

Line 141-259

I believe each therapy is well summarized.

Line 261-413

I think the main core part of this research is well organized. Especially, the sections that are very useful for practical application when actually applying the therapy are impressively well-compiled.

Overall, the organization of the results is commendable; however, procedural transparency is also necessary for the paper to hold value as a scholarly article. Therefore, I believe that if the research methodology section is elaborated upon, this study would possess significant value as a publication.

Methodology sect presented in a more detailed fashion in regards to search strategy and inclusion criteria. Aims of study clarified.

Reviewer 2 Report (New Reviewer)

Comments and Suggestions for Authors

Dear Editor,

Thank you very much for the opportunity to review this manuscript. This manuscript reports the results of a literature review regarding horticultural therapy and its practice recommendations for individuals with dementia. The manuscript is well written and presents practice recommendations in an understandable way.

We just have a couple of minor suggestions to enhance the clarity of this manuscript, as below:

1. In the abstract, there is no information that this is a review article. It would be good to inform the readers of the brief methodology of this review article.

2. Authors informed of the data sources and search strategy, including the number of articles yielded during the literature search on page 3. However, there is no information about how many articles were reviewed in this paper and no information about data analysis. Please provide information about how many articles were reviewed and how the reviewed articles were analyzed by the authors.

3. There are some writing errors and non-academic writing; for example, on page 8, “mix the soil the in closed bag,” and on page 12, “PLWD can’t always live …”. Please check the writing again.

Author Response

Reviewer 2

Dear Editor,

Thank you very much for the opportunity to review this manuscript. This manuscript reports the results of a literature review regarding horticultural therapy and its practice recommendations for individuals with dementia. The manuscript is well written and presents practice recommendations in an understandable way.

We just have a couple of minor suggestions to enhance the clarity of this manuscript, as below:

  1. In the abstract, there is no information that this is a review article. It would be good to inform the readers of the brief methodology of this review article. Abstract adjusted to reflect reviewer suggestion

  1. Authors informed of the data sources and search strategy, including the number of articles yielded during the literature search on page 3. However, there is no information about how many articles were reviewed in this paper and no information about data analysis. Please provide information about how many articles were reviewed and how the reviewed articles were analyzed by the authors. Search strategy detailed further with article counts and inclusion criteria for selection of cited articles-methodology section expanded to reflect more detail in search strategy and inclusion criteria

  1. There are some writing errors and non-academic writing; for example, on page 8, “mix the soil the in closed bag,” and on page 12, “PLWD can’t always live …”. Writing checked and corrected

Overall, the organization of the results is commendable; however, procedural transparency is also necessary for the paper to hold value as a scholarly article. Therefore, I believe that if the research methodology section is elaborated upon, this study would possess significant value as a publication.

Methodology sect presented in a more detailed fashion in regards to search strategy and inclusion criteria. Aims of study clarified.

Thank you for your time to review and offer feedback

Reviewer 3 Report (New Reviewer)

Comments and Suggestions for Authors

This paper explores an emergent issue of using horticultural therapy as a non-pharmacological intervention for individuals with dementia. This paper offers a foundational layer to the literature. While the paper provides some detail on variations of horticultural therapy implementation, it does not explore other aspects of the garden (e.g., herbs, fruits, propagation, etc.). Some of my comments can be found below:

1) Methods: Perhaps include 'horticultural therapy' in the themes or search strategy to expand studies that have used horticultural therapy as an intervention or treatment.

2) Results: To make your paper more robust, one suggestion is to include studies that have actually implemented these types of therapies on participants.

3) Variations of HT implementation: Consider expanding variations of HT implementation by including sessions that explore the lifecycle of plants through propagation or producing herbal bags (for sensory therapy as well). When handling soil is the only option involved, it can be overwhelming to participants--both for the participant with dementia and for their caregivers.

Author Response

Reviewer 3-This paper explores an emergent issue of using horticultural therapy as a non-pharmacological intervention for individuals with dementia. This paper offers a foundational layer to the literature. While the paper provides some detail on variations of horticultural therapy implementation, it does not explore other aspects of the garden (e.g., herbs, fruits, propagation, etc.). Some of my comments can be found below:

1) Methods: Perhaps include 'horticultural therapy' in the themes or search strategy to expand studies that have used horticultural therapy as an intervention or treatment.

Horticultural therapy added to search methodology. Methodology detail expanded to allow replication, inclusion criteria and aims of study included

2) Results: To make your paper more robust, one suggestion is to include studies that have actually implemented these types of therapies on participants.

Several review articles were used that detailed multiple studies and practical methods of the implementation of these types of therapies on participants, specifically for the purposes of BPSD.  This author thought that systematic reviews and meta-analyses offered the largest breadth of possible options and uses of different therapies for BPSD management in dementia care along with detailed activities provided in several tables along with their specific results according to type of intervention.  These articles include: Reference number 4, “Horticultural therapy in patients with dementia: A systematic review and meta-analysis” with the purpose of investigating “psychological health benefits of horticulture intervention in dementia patients”, reference number 8, “Therapeutic Horticulture for Dementia: A Systematic Review” which details “the evidence concerning the use of therapeutic horticulture (TH) as a non-pharmacological tool for decreasing neuro-psychiatric symptoms, such as agitation and depression, in PLWD”, reference number 16 & 17 , “A systematic review of Montessori-based activities for persons with dementia” and “Implementing Montessori methods for dementia: A scoping review” that found constructive engagement and positive affect improvement in PLWD, and lastly, reference number 30 “Using a multisensory environment to decrease negative behaviors in clients with dementia” to name a few.

3) Variations of HT implementation: Consider expanding variations of HT implementation by including sessions that explore the lifecycle of plants through propagation or producing herbal bags (for sensory therapy as well). When handling soil is the only option involved, it can be overwhelming to participants--both for the participant with dementia and for their caregivers alternative nature-based projects suggested projects in manuscript line 184-5, 192-194. Lines262-269 present examples of additional horticulture-based activities often utilized in HT practice

Section 5.4, “Soil-less Activity Example for Immunological Concerns” outlines the use of a sensory herbal bag activity for immunocompromised persons but is not limited to use in the general dementia care or even  geriatric populations overall.  The final sentence in this section was added for reference to different uses and variations.

Likewise, tables 2 & 3 were designed to outline general information needed to undertake any soil related activities such as propagation, transplanting or starting seed.  The example of a transplanting activity was specifically used in this article for the purpose of brevity and for its high success rate in such populations.  Another line was added at the end of this section for clarification of such options.

This manuscript is a resubmission of an earlier submission. The following is a list of the peer review reports and author responses from that submission.

Round 1

Reviewer 1 Report

Comments and Suggestions for Authors

Being a gardener as well as a clinician, I should welcome work on Horticultural Therapy and Dementia.  Unfortunately, this piece does better at confusing than enlightening readers.

ACADEMIC RESEARCH Issues

1)  Readers never learn how this research advances what is already known, nor how it documents or validates any new information (?) it introduces.  

This article cites smatterings of research, bouncing from Montessori to Mindfulness, from reminiscence therapy to mouse models for aromatherapy, but conspicuously avoids any reference to THE central journal and textbooks in this field.  The AHTA has published their Journal of Therapeutic Horticulture for many years now—why not publish there—or at least cite their work?

25 years ago, Wells’ Horticultural Therapy and the Older Adult Population, and
Simson & Straus’ Horticulture as Therapy, already proposed most of what is suggested here.  Since 2006 or 2007, Haller & Capra’s Horticultural Therapy Methods, Chalfont’s Design for Nature in Dementia Care, and subsequently Marcus & Sachs’ Therapeutic Landscapes, An Evidence-Based Approach have become the standard textbooks in this field, but (A) none of these classics are cited here, and (B) we are never shown how this article advances what was already proposed and researched years ago.

2)  The focus is scattered.  Is this about Horticultural Therapy, or HT and sensory reminiscence therapy, or HT and Montessori?  How does Montessori-Based HT differ from non-Montessori-Based HT?  Is this a red herring, or why should readers even care whether Montessori is related?  It needs definition, delimiting and focus.

LOGICAL  Issues
Many sentences and ideas are confusing if not self-contradictory.  A few cases:

(30-31)  “Such (Economic) numbers, spiking at a rapid rate (tautological; spike=rapid), leads to a rise in elderly individuals with dementia….”

>  Au contraire, it is the rise in elderly individuals with dementia that leads to global and personal costs, not the costs that lead to a rise in PWLD.

(46-48)  “BPSD can also influence other domains of function.  For example, symptoms may then cause…a lack of motivation/apathy”

>  BPSD symptoms do not CAUSE apathy; BPSD INCLUDES apathy already.

(36-40)  Neuropsychiatric symptoms…cause adverse outcomes for PWLD…including increased distress among patients and caregivers….”
>  first we are told the adverse outcomes are for PWLD, but then the adverse outcomes suddenly switch to caregivers’ distress.

(73-79)  “The Montessori Method relies heavily on agency/independence…to help treat BPSD…Therefore, a feeling of agency and independence becomes important.”

>  It is counterintuitive that agency/independence reduce disinhibition, delusion, euphoria, or wandering—such BPSD all show too MUCH agency/independence, not too little!  Readers need careful reasoning and evidence that more agency/independence will reduce disinhibition, delusion, euphoria, or wandering, not a simple “therefore” which non-causally reasserts its premise.

(157)  (Gardens) should contain several elements (like what?) while avoiding elements…   

(163)  The ability to go outdoors, especially alone, if/when appropriate….

>   This begs the question.  WHEN is it appropriate for BPSD patients to go out alone?   Assessment?  Level?

(175)   Most people from this age   (WHAT age??  Is dementia an age issue?)

(177)  (stimulation)  is often a must

>  “often” contradicts “mandatory”  Is it one or the other, and how to assess?

(198)   offer two or three choices (but not more)  VS.  (226)  offering four different herbs

>  What grounds for two or three choices (but not more) vs. four choices ?

(227)  (should offer….) lemon balm  VS.  (271/3)  precautions against lemon balm for adverse thyroid effects       

>   What grounds for using or not using lemon balm?  

(271)    Use potting mix/adding perlite   vs.  use materials over 1.5” to avoid choking

The left column of Table 6 shows (many non-BPSD) patient behaviors, while the right column gives recommendations for supervisor behaviors.  This difference in subjects should be clarified in headings.

This is an incomplete list, but such confusions/contradictions are inappropriate in our academic journal.

GRAMMATICAL Issues

Some “sentences” lack subjects or verbs.

Commas abound improperly between phrases, but are notably lacking between independent clauses of compound sentences.

Tables jump from nouns to phrases to gerunds to imperatives; these need unification and consistency.

In short, IF this article is revised to

(1)  show how it researches and validates its advances on what we already know;

(2)  avoid confusing and contradictory statements;  and

(3)  adopt standard English grammar

It might be worth another serious review for our journal.  As it stands, it is not ready for MDPI Healthcare.

Comments on the Quality of English Language

Some “sentences” lack subjects or verbs.

Commas abound improperly between phrases, but are notably lacking between independent clauses of compound sentences.

Tables jump from nouns to phrases to gerunds to imperatives; these need unification and consistency.

Author Response

Thank you for your detailed response and feedback. Points addressed in attached file.

Reviewer 2 Report

Comments and Suggestions for Authors

Confusing, There is no structure in accordance with any type of article in the scientific literature. No results section. 

Introduction: Dementia is mentioned, but it is not specified what type of dementia is the focus of the intervention. In the intervention itself, several non-pharmacological therapies are mentioned (without any previous literature review) but then only horticultural therapy is mentioned. Why are the others not included? What were the scientific criteria for choosing horticultural therapy?

Methodology: There is no definition of the type of study. Again, introductory considerations of horticultural therapy are mentioned, but there are no sections on methodology. Sample, type of study, assessment instruments... all the sections that are included in the methodology of the study are missing. It could be considered a narrative review, but what model did you use to conduct the review? What guidelines have been considered?

Results: There are no results

Discussion: Again descriptive, no results are discussed because they do not exist previously.

Author Response

Thank you for reviewing this manuscript. Responses to commentary in attached file.

Reviewer 3 Report

Comments and Suggestions for Authors

Congratulations to the authors, I found your article very interesting and of great interest.

I would like to be able to ask the authors whether this type of intervention has shown significant beneficts on global cognition and also wheter it has show significant beneficts on cognitive functions.

If so, I recommended extending the Discussion section and commenting on it in Conclusions section.  

Author Response

Thank you for reviewing this manuscript. Comments addressed in body of paper.

Reviewer 4 Report

Comments and Suggestions for Authors

The paper addresses the topic of non-pharmacological therapies for individuals with dementia.

The abstract is quite effective although the line of reasoning proposed in the paper is not fully paralleled. It starts with an analysis of several therapies before going to the target one.

Introduction

Pag 2, paragraph 1.1 A more detailed description of the MBP.

Page 3_ 1.4. This paragraph should also include a more detailed description of the results that can also motivate the suggestions and recommendations that follow.

Page 4_ Paragraph 2. The title of this paragraph (Method…) orients readers’ expectations to something different from what follows in the paper, which is not research data but a review of diverse implementations’ details.

Page 7, paragraph 2.3.  As titles of the previous paragraphs refer to the meaning of the analysis, the title here should refer to the content addressed more than mentioning the activity it refers to. Additionally, it is not proposed in the same modality of the previous paragraphs, namely with a table summarizing main contribution to the area.

Discussion

The authors dedicate a large space to details and suggestions for conducting the Horticultural therapy in the corpus of the paper and only afterwards, during the discussion, to describing reasons for proposing it. Readers may better understand the meaning and the relevance of the suggestions provided if they could appreciate in advance the relevance and impact of the therapy as emerging from the research studies.

So, I suggest reversing the order of presentation of the main topics and address in the discussion issues and proposal related to effective, person-centred, non-pharmacological interventions. 

Comments on the Quality of English Language

No specific comment at this level

Author Response

Tghank you for your time and constructive comments. Changes shown in attached file and in paper

Round 2

Reviewer 1 Report

Comments and Suggestions for Authors

The authors still provide no clear explanation of the points on which their research advances what is already known, nor the grounding and methods of their research, yet they have worked hard to upgrade their original draft to a publishable standard, so further critiques may be left to our readers.

Author Response

In regards to the comments:

This paper reflects current status and updates in the literature.

methods further clarified and adjusted in structure of text

Reviewer 2 Report

Comments and Suggestions for Authors

I strongly advise you to study what a review is. 

Selçuk, A. A. (2019). A guide for systematic reviews: PRISMA. Turkish archives of otorhinolaryngology, 57(1), 57.

In this case, methodological principles are not applied according to any existing methodology with scientific rigour.

It is important to understand that the introduction has improved, but a methodology that follows standardised methodological principles is still lacking and there are no results or discussion.

Author Response

Structure and methodology of paper modified to reflect a Narrative review. This format is supportive of the relatively narrow scope of the paper along with the addition of practice recommendations which are not always included in review papers. Further research recommended in conclusion. 

Reviewer 4 Report

Comments and Suggestions for Authors

Authors largely improved the paper, both in content and in structure. 

Although clearly it is orientd towards providing recommendations and suggestions for practice, I belive it is important readers are provided with details on the effects  and evidence from studies described in  the literature. Details of this sort  might be added at the bottom of paragraph 2.4, as the authors did  in the previous paragraphs.

Comments on the Quality of English Language

No particular comment

Author Response

Details regarding benefits and support in the literature provided in last paragraph of section.